# Effect of Chemical Treatments on the Properties of High-Density Luffa Mattress Filling Materials

**DOI:** 10.3390/ma12111796

**Published:** 2019-06-03

**Authors:** Kaiting Zhang, Yong Guo, Fangcheng Yuan, Tingting Zhang, Zhenzhen Zhu, Beibei Weng, ShanShan Wu, Tong Chen, Yuxia Chen

**Affiliations:** College of Forest and Garden, Anhui Agricultural University, Hefei 230036, China; 15905690096@163.com (K.Z.); fly828828@163.com (Y.G.); 18158860792@163.com (F.Y.); 18715068105@163.com (T.Z.); 18715063564@163.com (Z.Z.); weng1942@163.com (B.W.); 18297556933@163.com (S.W.); 18356092289@163.com (T.C.)

**Keywords:** softening treatment, cylindrical luffa mattress filling materials, wettability, firmness, compression resilience

## Abstract

Luffa is a lightweight porous material with excellent biocompatibility and abundant resources. In this paper, three kinds of softening treatment methods, alkali-hydrogen peroxide (Method 1), alkali-acetic acid (Method 2), and alkali-urea (Method 3), were used to soften high-density (HD) cylindrical luffa (CL) mattress-filling materials (MFM). Microscopic observation, mechanical performance testing and other analyses were performed to evaluate the effects of the three kinds of softening methods on the wettability, compression resilience and support performance of CL MFM. The results showed that: (1) After the treatment by Method 1, Method 2 and Method 3, the peak stress of CL decreased by 73%, 10% and 27%, respectively. In addition, after three kinds of softening treatments, the uniformity of CL increased. (2) When the CL MFM of high density rank treated by Method 1 was compressed by 40%, the firmness values of the surface, core and bottom reduced by 53.49% 40.72%, and 46.17%, respectively, compared to that of untreated CL. In addition, for the CL MFM of high density rank treated by Method 3 and then compressed to 60%, the firmness of the surface layer, core layer and bottom layer reduced by 41.2%, 33.7%, and 36.9%, respectively. (3) The contact angle of luffa treated by Method 3 was the smallest, next came Method 1 and Method 2, and untreated was the largest. (4) After the treatment by Method 3, the fiber bundle of luffa was intact, and the compression resilience of the CL was obviously increased. Therefore, this method can effectively reduce the firmness of MFM and also improve the uniformity and wettability of CL.

## 1. Introduction

Sleep is essential to guarantee energy, physical recovery and good health, and people spend a third of their life in sleep [1]. The mattress is the main furniture for sleeping, and the performance of mattress-filling materials (MFM) is an important aspect that affects comfort and quality of sleep [2]. The firmness of MFM has a significant impact on the sleep quality of healthy people. The effect of mattress firmness on the human body includes two aspects, one is support stability and the other is human–bed interface pressure distribution [3]. In addition to the firmness of the MFM, the moisture absorption and dissipation properties, air permeability and environmental friendliness also affect the performance of the mattress. Humidity control of a mattress is the ability of a mattress to absorb and dissipate moisture, and it is an important performance factor that affects the sleep microenvironment [4]. According to surveys, the human body emits water (sweat) and vapor, equivalent to 200–300 mL of water, to the surrounding environment through breathing and body surface every night. The moisture emitted by the human body surface is absorbed by mattresses, bedding and pillows [5]. If the moisture absorbed by the mattress cannot be dissipated as soon as possible, it would lead to accelerated rusting of mattress springs and growth of mildew on the materials. This in turn would affect the service life of the mattress and human health. In addition, the absorbed moisture can make the mattress surface moist and sticky, which would stimulate the skin and make the human skin rough and sensitive [6]. In 2017, Chen et al. found that due to the existence of porous structure and surface grooves, the moisture retention of the luffa fiber bundles was superior to that of jute and coconut palm [7]. However, the moisture regain of the luffa decreased because the surface topography changed after alkali treatment [8]. The moisture absorption properties of plant fibers include water absorption and wettability, and these properties are mainly determined by the physical and chemical structure of the plant fibers. Therefore, further research is needed on whether the moisture and humidity control of the cylindrical luffas (CL) MFM after the softening treatment can satisfy the requirements of the mattress.

The plant fiber mattresses (jute and palm mattress) on the market now have high firmness, poor air permeability and low moisture desorption properties [7], affecting the healthy sleep environment. Luffa is a porous material with a multi-level pore structure. As luffa has a fibro-vascular reticulated structure [7,9], extremely high porosity [10] and high air permeability [11], it can absorb water more than 2.0–3.5 times its own weight and has a moisture dissipation rate (13 h) as high as 76.86–91.44% [7,9]. Compared with low-density luffa, high-density (HD) luffa has more abundant resources and better dimensional stability. However, the compressive strength (0.15 MPa–0.4 MPa) and peak stress (0.07 MPa–0.3 MPa) of HD luffa with the density range of 31 kg m^−3^–65 kg m^−3^ are about 10 times higher than the corresponding values for low-density (15 kg m^−3^–35 kg m^−3^) luffa [9]. Thus, the firmness of the MFM prepared by HD luffa is much higher than that of the MFM prepared by low-density luffa. Therefore, it is necessary to reduce the firmness of CL MFM by chemical treatment, so that it is suitable to be used for the preparation of soft and comfortable luffa mattresses.

Alkali treatment is the main method for softening plant fibers. In 2018, Chen et al. used three chemical treatment methods (5% NaOH-5% H_2_O_2_, 10% NaOH-20% CH_3_COOH and 18% NaOH-1.6% CO(NH_2_)_2_), to modify the luffa fiber bundles in middle layer and found that all three methods can partially remove hemicellulose, lignin and other amorphous substances. These treatments resulted in the formation of nanopores between the well-arranged cellulose nanofibers, which improved the softening and flexibility of the cell walls of the fibers [12]. In addition, the alkali treatment caused the fibers to swell and destroyed the hydrogen bonds between the macromolecular chains. As a result, the elastic modulus of the fiber bundles was lowered and the luffa fiber bundles were softened [13]. In 1962, Chakravarty used NaOH followed by acetic acid (CH_3_COOH) to treat jute fiber. The results showed that 10–18% NaOH and 15% CH_3_COOH-treated jute fiber acquired the finest elasticity and flexibility [14]. In 2017, Mittal investigated into the influence of alkaline hydrogen peroxide pretreatment on delignification. The results showed that the pretreated solids with more than 80% delignification consequently enriched the carbohydrate fraction to >90% [15]. In 2009, Ghali et al. studied the effect of alkali treatment and alkaline hydrogen peroxide on mat structure of luffa fibres. The results showed that both treatments resulted in a removal of lignin, pectin and hemicellulose substances, and changed the characteristics of the surface topography [8]. Xiong’s group performed a very intensive study on the dissolution mechanism of cellulose in NaOH/urea solvent. They found upon the addition of a certain amount of urea or thiourea to NaOH solution, the solution penetrates into the crystal region of cellulose, where it destroys the hydrogen bonds, leading to changes in the aggregate structure of the crystalline region of cellulose, and reduces the crystallinity [16].

Luffa is a kind of anisotropic and heterogeneous material with different arrangement modes in four layers of CL. Therefore, its mechanical properties are not only related to luffa fiber bundles, but are also related to the arrangement mode and the network structure [7,9,17]. It is known that the mechanical properties of CL improve with the increase in density [9,18]. In order to obtain MFM with uniform support performance, the CL is first classified and pre-compressed, and then CL elastomer is quilted by non-woven fabric. After compression treatment, the surface of the luffa fiber bundle may have folds and cracks [9], which would affect the firmness and resilience of CL MFM. Moreover, after the CL is compressed, it rebounds under internal stress. An appropriate amount of rebound can ensure the elasticity, stability and comfort of the mattress. Therefore, further research is needed on whether the alkali-treated CL can be used for preparing MFM and the effect of alkali treatment on the support properties of MFM.

In the research of luffa, most scholars focus on the performance of luffa as a composite material and packaging material; only a few scholars study the performance of luffa as a mattress cushioning material. Our previous research has shown that natural high-density cylindrical luffa has the potential to become a mattress cushioning material, but its firmness was large. At present, there is no research on the softening of cylindrical luffa, and the research about the influence of chemical treatment on the mechanical properties of MFM also has gaps.

In this paper, three softening methods (5% NaOH-5% H_2_O_2_, 10% NaOH-20% CH_3_COOH and 18% NaOH-1.6% CO(NH_2_)_2_) were used to treat HD CL. The effects of these three softening treatments on the properties of CL MFM were studied by compression resilience test, wettability test, firmness, and hysteresis loss rate test.

## 2. Materials and Methods

### 2.1. Sample Preparation

#### 2.1.1. Luffa Materials

In this study, the producing area of HD luffa was Shitai, Anhui in China. The aging parts of the luffa at the two ends were removed by a circular saw (CMS-TS-55-R Set, Festool, Xiamen Hengkaite Industry and Trade Co., Ltd., Xiamen, China), and the middle part was sawed into 50 mm cylinders. The cut cylinders were washed with water and dried, and then placed in a humidity chamber (HWS-250C, Hefei Huadeli Scientific Equipment Co., Ltd., Hefei, China) with 65% relative humidity (RH) at 21 °C for 24 h. Finally, the density and volume of CL were calculated [9].

The density of luffa cylinders was calculated by using Equation (1):(1)$$\rho =\frac{2m}{\left(S_{1}+S_{2}\right)\times H}$$where *ρ* is the density of the luffa cylinder (kg/m^3^), *m* is the mass of the luffa cylinder (kg), and *S*_1_ and *S*_2_ are the areas of the upper and bottom cross sections of the luffa cylinder excluding the pores on the cross sections (m^2^), respectively. *H* is the average of 10 measurements at different points of the luffa cylinder (mm). Figure 1 shows the locations of the 10 height test points of the luffa cylinder.

Specific steps of measuring *S*_1_ and *S*_2_ are as follows: First, mark two points on the upper and lower surface of the cylindrical luffa land and then measure the distance between them *L*_1_ and *L*_2_. The straight line between the two points should pass through the upper or lower surface of the luffa. Using a digital camera (IXUS120, Henan, China) to shoot the upper and lower surface of the cylindrical luffa, the shooting needs to be parallel to the surface of the cylindrical luffa. Then, import the two photos into CAD software and trace the surface of the luffa with ‘SPL’ command, as shown in Figure 1. Then use the ‘H’ command to fill the enclosed part, and use the ‘AREA’ command to click the filled part to get the CAD area of *A*_1_ and *A*_2_. Finally, the distance between the two markers in the picture *X*_1_ and *X*_2_ are measured by the straight line. The surface area *S*_1_ and *S*_2_ of the cylindrical luffa can be calculated according to the following formula:(2)$$S_{1\;or\;2}=\frac{L_{1\;or\;2}{}^{2}}{X_{1\;or\;2}{}^{2}}\times A_{1\;or\;2}$$

#### 2.1.2. Three Chemical Treatment Methods

After testing the density, the HD CL samples were divided into three density ranks (Low: 30–41 kg m^−3^, Medium: 42–50 kg m^−3^, and High: 52–65 kg m^−3^) according to the density. Then, each density rank was randomly divided into four groups. One group was untreated and the other three groups were treated with 5% NaOH-5% H_2_O_2_ (Method 1), 10% NaOH-20% CH_3_COOH (Method 2) and 18% NaOH-1.6% CO(NH_2_) (Method 3), respectively. The steps and processing parameters of the three softening treatments refer to a previous publication [13]. 

Treated CL samples were rinsed with distilled water several times and dried at room temperature for 5 h. Then, they were placed in a constant temperature and humidity oven for 24 h, after which the density and volume of the resulting CL samples were calculated.

#### 2.1.3. Sample Preparation

The preparation process of the CL MFM sample is described below (taking the untreated luffa as an example): The high, medium and low density rank samples of untreated CL were randomly divided into two groups. Then, the two groups of CL were compressed by 40% and 60%, respectively, along the height direction to obtain six kinds of CL elastomers. The compression conditions (XLB300, Qingdao Xincheng Yiming Rubber Machinery Co. Ltd., Qingdao, China) were: hot pressing time of 10 min, temperature of 50 °C and pressure of 0.1 MPa. Then, the CL elastomer was quilted by non-woven fabric into a 300 mm × 400 mm pad. Multi-layer luffa cushions were stacked into the fabric bag to prepare the CL mattress specimens, as shown in Figure 2. Three duplicate specimens were obtained. The process of softening CL MFM was similar to that of untreated luffa. 

The CL samples used for testing the water contact angle properties were cut alongside the hoop wall and flattened first, and then trimmed into a sheet of 30 mm × 30 mm. Five duplicate specimens were obtained.

### 2.2. Quasi-Static Uniaxial Compression

First, the CL sample was placed in an environment with temperature of 20 °C and RH of 65%. After 24 h, the CL was taken out and placed under the pressure head of a mechanical test machine (Shimadzu Corporation, Shimadzu AG-X Plus, Kyoto, Japan). The crosshead speed was 3 mm min^−1^. The maximum compression of the CL was 80%. CL were tested at room temperature of 25 °C and RH of approximately 50%. The plateau stress can be obtained by calculating energy dissipation efficiency (Figure 3), using the following formulas:(3)$$E_{d}\left(\varepsilon_{a}\right)=\frac{\int_{0}^{\varepsilon_{a}}\sigma \left(\varepsilon \right)d\varepsilon }{\sigma_{a}},\;0\leq \varepsilon_{a}\leq 1$$
(4)$$\frac{dE_{d}\left(\varepsilon_{a}\right)}{d\varepsilon }/\varepsilon_{a}=\varepsilon_{i}=0,\;0\leq \varepsilon_{i}\leq 1$$
(5)$$\sigma_{pl}=\frac{\int_{0}^{\varepsilon_{d}}\sigma \left(\varepsilon \right)d\varepsilon }{\varepsilon_{d}}$$
where $E_{d}$ is the energy dissipation efficiency, $\varepsilon_{i}$ is the strain, $\varepsilon_{d}$ is the densification strain defined as the maximum value of $\varepsilon_{i}$, and $\sigma_{pl}$ is the plateau stress.

### 2.3. Morphology Observation of LS Fiber Bundles by Scanning Electron Microscopy (SEM)

The morphologies of the transverse surface of luffa samples were observed using a Hitachi SEM (HitachiS-4800 microscope, Tokyo, Japan) with an acceleration voltage of 5 kV. The samples were sputter-coated with gold prior to observation after natural drying (temperature of 20 °C and RH of 65%) [19].

### 2.4. Basic Mechanical Characteristics of Mattress Filled with Columned Luffa

Before the experiment, the MFM was placed in an environment with a temperature of 20 °C and RH of 65% for 24 h. A mechanical test machine (Shimadzu Corporation, Shimadzu AG-X Plus, Kyoto, Japan) was used for measuring the mechanical properties of MFM. The diameter of the upper pressure head was 100 mm, and the loading speed was 100 mm min^−1^. Figure 4 shows the method for calculating the mechanical characteristics of the mattress filled.

The tests of firmness and hysteresis loss rate of the MFM were performed according to the standard ISO 2439:2008 (E). The firmness of the mattress specimens is given by *D_surface_*, *D_core_* and *D_bottom_*, which can be calculated by the following Equations (6), (7) and (8), respectively:(6)$$D_{surface}=\frac{36/S}{\varepsilon_{f40N}-\varepsilon_{f4N}}\times 10^{-6}$$
(7)$$D_{core}=\frac{160/S}{\varepsilon_{f200N}-\varepsilon_{f40N}}\times 10^{-6}$$
(8)$$D_{bottom}=\frac{50/S}{\varepsilon_{f250N}-\varepsilon_{f200N}}\times 10^{-6}$$
(9)$$H_{hysteresis}=\frac{A_{1}}{A_{2}}\times 100\%$$
where $D_{surface}$, $D_{core}$ and $D_{bottom}$ (in MPa) represent the modulus between 4 and 40 N (surface), 40 and 200 N (core), and 200 N and 250 N (bottom), respectively. The terms $\varepsilon_{f4N}$,$\;\varepsilon_{f40N}$, $\varepsilon_{f200N}$ and $\varepsilon_{f250N}$ represent the strain at 4 N, 40 N, 200 N and 250 N, respectively. The term *S* (in m^2^) represents the area of the pressure head; $A_{1}$ represents the area between the load curve and the relief curve (both limited to 250 N), and $A_{2}$ is the area under the load curve (limited to 250 N). For mattresses, the higher the values of $D_{surface}$, $D_{core}$ and $D_{bottom}$, the softer their support performance is.

The test of the compressive deflection coefficient was conducted according to the standard of ISO 2439:2008 (E). First, the specimen was loaded to (75 ± 1) % of thickness, and then unloaded. The compressive deflection coefficient, *S_f_*, was calculated as follows:(10)$$S_{f}=\frac{F_{65}}{F_{25}}$$where *F*_25_ is the force at 25% indentation in compression (in N), and *F*_65_ is the force at 65% indentation in compression (in N).

### 2.5. Compression Resilience

The compressed LS columns (20 and 30 mm in height and 60% and 40% in compression) were placed in a humidity chamber at 20 C and 65% RH for 24 h. Then, the changes in the height of CL were recorded before and after constant temperature and humidity treatment. The calculation was performed using the following formula:(11)$$h=\frac{H_{2}-H_{1}}{50}\times 100\%$$
where *h* is the percentage of compression resilience of columned luffa, and *H*_1_ and *H*_2_ are the heights (in mm) of columned luffa before and after constant temperature 20 °C and humility 65% treatment for 24 h. Both these values were the mean of 10 testing points.

### 2.6. Water Contact Angle Measurement

The measurement of contact angle was carried out using an automatic contact angle meter (Model: CA100D, provided by Yingnuo Shanghai Instrument Co., Ltd., Shanghai, China). Before testing, the sample was pressed into a flat surface by a tablet press (YP-2, Shanyue Shanghai Instrument Co., Ltd., Shanghai, China). During the contact angle measurement process, the sample was first placed on the platform and a fixed volume (∼2 μL) of deionized water was injected on the sample surface using a syringe. Then, the images of the water droplet were taken using a CCD camera (IXUS120, Henan, China), as shown in Figure 5. Finally, these images were analyzed using the supplied software to determine the contact angle [20]. Each kind of sample was measured five times and averaged as the sample contact angle.

## 3. Results and Discussion

### 3.1. Density and Plateau Stress of Treated and Untreated Cylindrical Luffa

Figure 6 presents the density and plateau stress values of the HD CL after the three alkali treatments. As shown in Figure 6a, the density ranges of CL treated with Method 1 were the same as those of the untreated CL, which was 30–65 Kg m^−3^. On the other hand, the density of CL samples treated by Methods 2 and 3 increased, and the density ranges were 38–90 Kg m^−3^ and 37–88 Kg m^−3^, respectively. Table 1 shows the correlation between the density of CL before and after softening treatment. It can be seen from Table 1 that there was a significant correlation of the density of CL samples after and before treatment. The corresponding correlation coefficients for Method 1, Method 2 and Method 3 treated samples were 0.92, 0.708 and 0.808, respectively. All three values were above 0.7, which indicates that CL after softening treatment can be classified by the density of CL before softening treatment to ensure the uniformity of the support performance of the mattress (Low: 30–41 kg m^−3^, Middle: 42–50 kg m^−3^, High: 51–65 kg m^−3^). Among the three softening methods, the volume shrinkage of CL was the smallest and the mass loss was largest after the treatment by Method 1, while the mass loss rate of the luffa was smaller and the volume shrinkage was larger after the treatments by Method 2 and Method 3 [13]. Therefore, there was less change in the density range after Method 1 treatment, while the density increased after the treatment by Method 2 and Method 3.

Table 2 shows the plateau stress and its fitting parameters of CL. Plateau stress is the ratio of energy dissipation at the condition of the maximum efficiency to densification strain. It is an important index to characterize the energy absorption capacity of materials [21]. Its value is similar to compressive strength, so it can indicate energy absorption capacity and deformation capacity of CL. According to Table 2 and Figure 6b, the peak stress of untreated CL was 0.175 ± 0.069 MPa, and it decreased by 73% after the treatment by Method 1, which was 0.047 ± 0.021 MPa and larger than that of the low density CL (0.01–0.035 MPa) [9]. Thus, it can be seen that the CL treated by Method 1 still had good energy absorption characteristics. Method 1 can dissolve the pectin of binding fiber cells distributed in the middle layer of cells, leading to the exposure and separation of the fibers [22,23]. Moreover, Method 1 can oxidize lignin, resulting in the formation of new carboxyl groups that are hydrophilic [24,25,26] and enable the macromolecules to degrade into small fragments. This facilitates the dissolution of lignin in water [27]. Figure 7 shows the surface morphology and microstructures of untreated and softened luffa fiber bundles after compression. It can be seen from the uncompressed luffa in Figure 7b, that after the treatment by Method 1, the surface of luffa fiber displayed a large number of grooves, pores and cracks. Moreover, the cell gaps showed many cracks, and the cells collapsed and deformed. In addition, the cell wall was obviously thinned and some cells were separated from each other. Consequently, the mechanical properties of CL were significantly reduced. As shown in Figure 7c,d, Method 2 and Method 3 also caused a large number of grooves and shrinkages and a few pores to appear on the surface of luffa fiber. Also, the fiber cell wall shrank and evolved from open latticed hexagonal cell lumina to a crumpled structure with irregular shape that showed excellent elasticity [12]. Therefore, the mechanical properties of CL were slightly decreased after the treatment by Method 2 and Method 3. The plateau stress of CL samples treated with Methods 2 and 3 decreased by 10% and 27% to 0.157 ± 0.073 MPa and 0.127 ± 0.058 MPa, respectively. 

From the fitting results in Table 2, the fitting exponents for the untreated CL, Method 1 treated CL, Method 2 treated CL and Method 3 treated CL were 2.03, 1.63, 1.86 and 1.87 respectively. These values were higher than that reported by Gibson et al. (1.5), but within the range of 1.5–3 found in other experimental work [21]. Similar to the compressive strength of CL, the exponent had fewer penalties for strength and energy absorption with decreasing density [18]. The exponent of CL decreased after softening treatments, and thus, the uniformity of CL increased.

According to previous studies, the densification strain of CL was independent of density, was only related to height [9], and the densification strain range was 0.6–1 [17]. In addition, the CL samples which were compressed to the plateau stage showed stable support performance [9]. Therefore, in this research, 50-mm-high CL samples were compressed by 40% (to the plateau stage) and 60% (to the densification stage). Then, these compressed samples were used for the preparation of MFM and to study the firmness, hysteresis loss rate, compression resilience, and wettability of MFM.

### 3.2. Basic Mechanical Characteristics of Mattress Filled with Columned Luffa

Table 3 and Table 4 show the mechanical property parameters of MFM when the CL was compressed by 40% and 60%. The results indicate that the firmness of MFM significantly decreased after softening treatment. For the MFM in which the CL was compressed by 40%, the firmness of MFM treated by Method 1 decreased most significantly, followed by those treated by Method 3 and Method 2. For the MFM in which the CL was compressed by 40%, the softening method which showed the greatest reduction in the firmness of MFM was Method 3. The three treatments led to the removal of non-cellulose substances such as lignin and hemicelluloses from luffa, which caused the appearance of many pores and grooves in the surface of luffa fiber bundles. This in turn decreased the deformation resistance capacity of CL, thus causing the decrease in firmness of MFM. 

After the treatment by Method 1, the non-cellulose substances in CL were largely removed, and the peak stress was reduced by 73%. After compression treatment, the inner fiber bundles of CL treated by Method 1 were bent and even broken. Thus, the deformation resistance capacity of fiber bundles reduced [9], resulting in a significant decrease in the firmness of MFM after the treatment by Method 1. The peak stress of CL was decreased by 10% and 27% after being treated by Methods 2 and 3, respectively. Among them, the Young’s modulus of luffa fiber bundle decreased by 61.7% and the content of luffa cellulose decreased most significantly after the treatment by Method 3. Most of the hemicellulose is located on the intercellular layer and cell walls [28]. Therefore, a decrease in the hemicellulose content resulted in less cell-to-cell binding. As shown in Figure 7d, there were many cracks on the surface of the inner luffa fiber bundle when the CL samples were compressed to the densification stage. Consequently, the structure of the luffa fiber bundle became loose and the strength of CL significantly decreased. Therefore, the firmness reduction of the CL MFM was more obvious after the treatment by Method 3. 

Furthermore, Table 3 and Table 4 show that the firmness values of untreated CL and treated CL MFM samples increased gradually from their top to bottom layers. This means that the mattresses had a softer surface layer and harder bottom layer. Moreover, the firmness of MFM in which the CL was compressed by 60% was significantly lower than that in which the CL was compressed by 40%.

As shown in Table 3, after the treatment by Method 1 and Method 3, the firmness of each layer of MFM in which the CL was compressed by 40% was reduced. The higher the density level, the greater the reduction in firmness was. Compared with the $D_{surface}$, $D_{core}$ and $D_{bottom}$ values (0.129 MPa, 0.334 MPa, 0.379 MPa) of A13, after the treatment by Method 1, the firmness of the three layers of B13 decreased by 53.49, 40.72, and 46.17%, respectively. The firmness of most layers of MFM after treatment by Method 2 was decreased. However, the $D_{core}$ and $D_{bottom}$ values of C12 were 0.167 MPa and 0.192 MPa, respectively, and increased by 15.60% and 38.00% compared to untreated CL. This was mainly because in Method 3-treated luffa, the non-cellulose substances were partially removed, the amount of compression of cell lumens of luffa fibers bundles was relatively small, and the cell walls were thick and intact. This led to the increase in elasticity of fiber bundles, and the fiber structure was more compact after compression, thereby causing an increase in the firmness of MFM. 

In terms of the hysteresis loss rate, the lower the density rank of CL, the greater the hysteresis loss rate of MFM. The hysteresis loss rate of MFM after Method 2 and Method 3 treatments was slightly smaller than that of the untreated sample. For the same density grade, the hysteresis loss rate of MFM treated by Method 1 was the largest, but it was less than that of low density CL MFM (67–75.7%). Many grooves, holes and cracks appeared on the surface and intercellular layer of luffa fibers after Method 1 treatment, and the porosity increased. Therefore, the energy absorption capacity was better during the compression process, as shown in Figure 6b.

It can be seen from Table 3 that the compression deflection coefficients of all MFM samples were more than 3.5. A high value of compression deflection coefficient indicates that the MFM has a comfortable touch, while providing good support [29]. Comparison of compression deflection coefficients of the three kinds of softened MFM showed that the value of MFM treated by Method 1 was the largest, followed by Method 2 and Method 3.

Table 4 shows the mechanical property parameters of MFM in which the CL was compressed by 60%. When compressed by 60%, the support performance of CL mainly depended on the bending resistance of luffa fiber bundles [17,30]. From Table 4, it was found that some of the MFM treated by Method 1 and Method 2 had higher firmness compared to the untreated sample. For example, the $D_{core}$ and $D_{bottom}$ values of MFM specimen C22 were 0.145 MPa and 0.231 MPa, respectively. These values were higher than those of A22 [$D_{core}\;$(0.137 MPa) and $D_{bottom}\;$(0.181 MPa)]. In addition, after the treatment by Method 3, the firmnesses of the MFM of all three density ranks were significantly reduced. Compared with A23, the $D_{surface}$, $D_{core}$ and $D_{bottom}$ values of D23 were decreased by 41.2%, 33.7% and 36.9%, respectively. These values were also less than those of A21, but greater than those of low density samples.

By comparing Table 3 and Table 4, it was found that the hysteresis loss rate of MFM in which the CL was compressed by 60% was similar to that of MFM in which the CL was compressed by 40%. However, the compression deflection coefficients of MFM in which the CL was compressed by 60% were significantly larger than that of MFM in which the CL was compressed by 40%. This indicates that the MFM in which the CL was compressed by 60% had good energy absorption characteristics. Moreover, it showed high support performance under high load.

From the above analysis, it is evident that the treatment by Method 1 can effectively reduce the firmness of CL MFM. However, after Method 1 treatment, many fractures appeared on the inner luffa fiber surface after compression treatment, which would affect the service life of the CL mattress. The firmness of CL MFM after Method 3 treatment was also significantly reduced. The firmness of the MFM in which the CL was compressed by 60% after Method 3 treatment was less than that of the MFM after Method 1 treatment. Furthermore, after the treatment by Method 1, the cell walls of luffa fiber were thick and intact, and the elasticity of the fibers increased. Therefore, the CL MFM treated by Method 3 was more suitable for mattress production.

### 3.3. The Compression Resilience of Untreated and Treated Luffa

The height of compressed CL will be restored to some extent after absorbing moisture from the air [31,32]. This value indicates the resilience of the material and determines the thickness of the MFM. Higher resilience value of CL provides better softness. However, the value of resilience cannot be too large, as it will affect the stability of MFM. Figure 8 shows the compression resilience of CL compressed by 40% and 60% after being kept in a humidity chamber at 20 °C and 65% relative humidity for 24 h. Figure 8 shows that after softening treatment, the compression resilience of CL was obviously increased compared to that of the untreated sample. This was because all three softening methods removed non-cellulose substances, increased relative content of cellulose [33], shrank the luffa fiber cells [13], and increased the elasticity of fiber bundles [12]. These changes in turn increased the compression resilience of CL. In addition, the moisture regain and internal stress of luffa fiber bundles also affected its compression resilience [34,35]. When the CL was compressed by 40%, the internal porous structures were partially collapsed. When the CL was compressed by 60%, the fibro-vascular reticulated structure of CL was compacted. Therefore, the deformation of fiber bundles was larger for 60% compressed CL compared to 40% compressed CL. From Figure 7, it can be seen that the fiber surface had many cracks when the CL was compressed by 60%. As a result, the luffa can absorb more water in wet air. Consequently, the compression resilience of CL compressed by 60% was larger than that of CL compressed by 40%. High compression resilience can ensure that the height of CL is restored distinctly after a period of time. Moreover, after compression, the structure of luffa fiber became looser [9], and the softness of CL increased after recovery. Method 2 removed less non-cellulosic substances from luffa, and consequently, the relative content of cellulose and the elasticity of fiber bundles increased correspondingly [13]. Therefore, the CL after treatment by Method 2 had greater compression resilience than that of CL treated by Method 3 and Method 1.

Comparing the compression resilience values of four kinds of CL compressed by 40%, it was found that the compression resilience of CL treated by Method 1 was the highest at 10.84%, and the resilience height was about 5.4 mm. The compression resilience of CL treated by Method 2 was the smallest at 7.68% and less than that treated by Method 3 (10%). Both these values were larger than 3.84 mm. 

When compressed by 60%, the CL treated by Method 3 had the largest compression resilience of 13.2%, followed by Method 1 (12%), Method 2 (9.7%) and untreated (4.2%) samples. The resilience heights of all sample groups were less than 6.6 mm. 

It was found from the above study that after softening treatment, the compression resilience of CL was obviously increased compared with untreated CL samples. After the treatment by Method 3, the compression resilience of CL, whether compressed by 40% or 60%, was larger, and the resilience height was 5.4–6.6 mm less than that of untreated CL. This result indicates that Method 3 can effectively improve the compression resilience of CL while ensuring dimensional stability. 

### 3.4. Wettability of Untreated and Treated Luffa

Water contact angles were measured to determine the surface wettability. Depending on the values of the contact angle, surface properties are determined as hydrophobic (contact angle > 90°) or hydrophilic (contact angle < 90°) [20]. The smaller the contact angle, the better the wettability. As shown in Table 5, after the softening treatments, the contact angle of luffa was reduced. Specifically, the contact angle of luffa treated by Method 3 was the smallest (57.1°). The contact angles for samples treated by Method 1 (59.0°) and Method 2 (60.9°) were higher, and the untreated luffa had the highest contact angle (74.1°). Many factors affect the contact angle, such as the number of polar hydroxyl groups on the surface [36], roughness, the number and size of pores, etc. [37]. From Figure 7, it can be seen that after softening, the waxy impurities on the surface of luffa fiber were removed, leading to exposure of the hydrophilic substances such as lignin and hemicellulose [13,36]. Also, a large number of voids and grooves appeared on the surface of the fiber bundles, resulting in the decrease in contact angle of luffa. This result shows that although the moisture regain of CL after softening treatment decreased, the wettability increased. Therefore, the luffa can absorb moisture generated by the human body quickly and ensure the dry-touch interface of the mattress. In addition, it can be seen from Figure 7 that after softening treatment, the number of cracks and pores on the surface and middle lamella increased, the cell cavity shrank, and the capillary tension was improved [38]. Furthermore, after softening treatments, the number of hydroxyl bonds in luffa fibers were decreased [13]. Due to these factors, the luffa after softening treatments had good moisture dissipation. 

## 4. Conclusions

In this study, three softening methods were used to treat HD CL, all of which were able to reduce peak stress and improve the uniformity of HD CL. Moreover, the three methods also changed the properties of wettability, compression resilience and support of the MFM. The following conclusions can be drawn from the results of this work: 

(1) After softening treatment, the non-cellulosic material in luffa was partially removed, and the relative content of cellulose was increased. The fiber cell wall shrank and evolved from an open latticed hexagonal cell lumina to a crumpled structure with irregular shape which showed excellent elasticity of luffa fiber. Thus, the compression resilience of CL after softening treatment was increased. Among them, the CL treated with 18% NaOH-1.6% CO(NH_2_)_2_ had the highest compression resilience. The resilience height was less than 6.6 mm, which satisfied the processing requirements of MFM. There was a significant correlation of the density of CL samples after and before treatment, which indicates that CL after softening treatment can be classified by density of CL before softening treatment to ensure the uniformity of the support performance of the mattress.

(2) All three softening treatments reduced the firmness of MFM. After treatment with 5%NaOH-5% H_2_O_2_, the firmness of MFM was effectively reduced. However, the structure of luffa fibers was damaged, which decreased the durability of the CL mattress. After treatment by the method of 18% NaOH-1.6% CO(NH_2_)_2_, the plateau stress of CL was reduced by 27%, and the MFM prepared using this CL compressed to the densification stage (compressed by 60%) had the lowest firmness. Moreover, the luffa fiber bundles were complete after treatment by this method. Therefore, in order to ensure the durability and softness of the mattress, treatment with 18% NaOH-1.6% CO(NH_2_)_2_ was the best softening method.

(3) After the softening treatment, the waxy impurities on the surface of luffa fiber were removed, which caused the hydrophilic substances such as lignin and hemicellulose to be exposed. As a result, the wettability of luffa increased, and it could absorb moisture generated by the human body quickly. Furthermore, the number of cracks and pores increased, the cell cavity shrank, and the capillary tension was improved in the treated luffa. Therefore, the luffa after softening treatments also had good wet dissipation. 

From the above research, the results show that the three chemical treatments can reduce the firmness and improve the compression resilience and wettability of the cylindrical luffa mattress-filling materials, broaden the market of raw materials for the development of plant fiber mattress, and provide a theoretical basis for the subsequent development of plant fiber mattress. Later, we will use the softened cylindrical luffa mattress-filling materials to develop the luffa mattress and study the influence of the cylindrical luffa on the comfort of the mattress.

## Figures and Tables

**Figure 1 materials-12-01796-f001:**
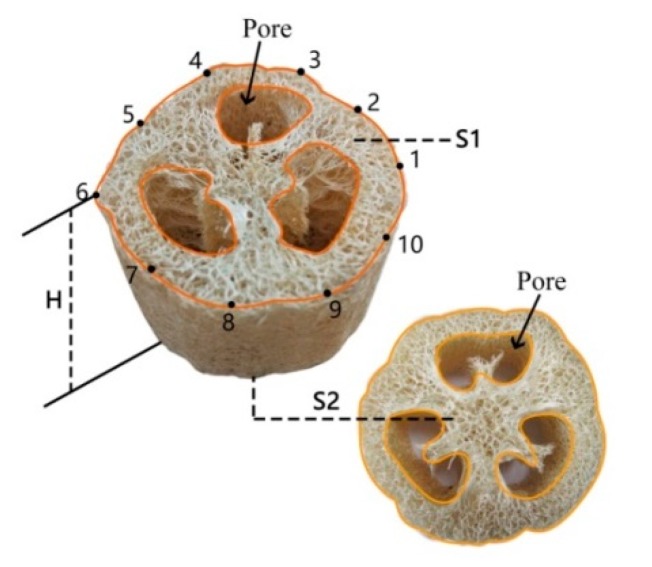
Locations of the 10 height test points of luffa cylinder.

**Figure 2 materials-12-01796-f002:**
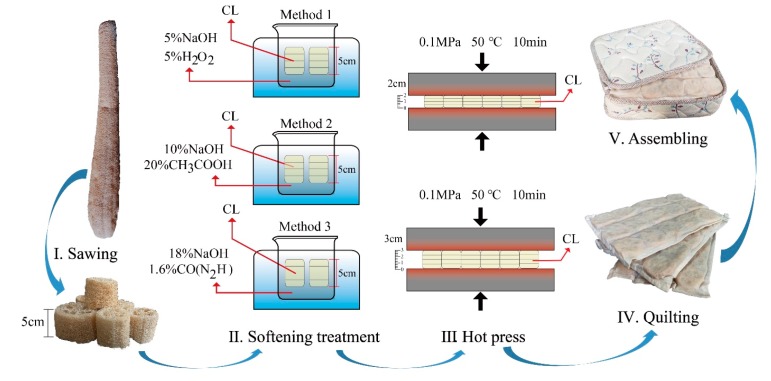
Preparation process of luffa filling material of mattress.

**Figure 3 materials-12-01796-f003:**
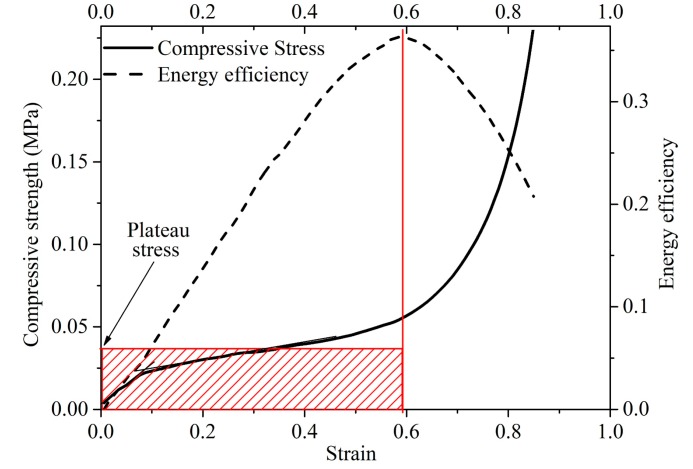
Illustration of energy efficiency method.

**Figure 4 materials-12-01796-f004:**
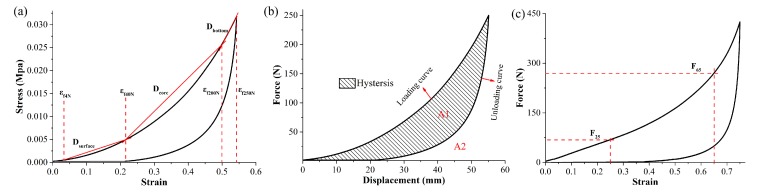
Illustration of calculating method of mechanical characteristics of mattress filled.

**Figure 5 materials-12-01796-f005:**
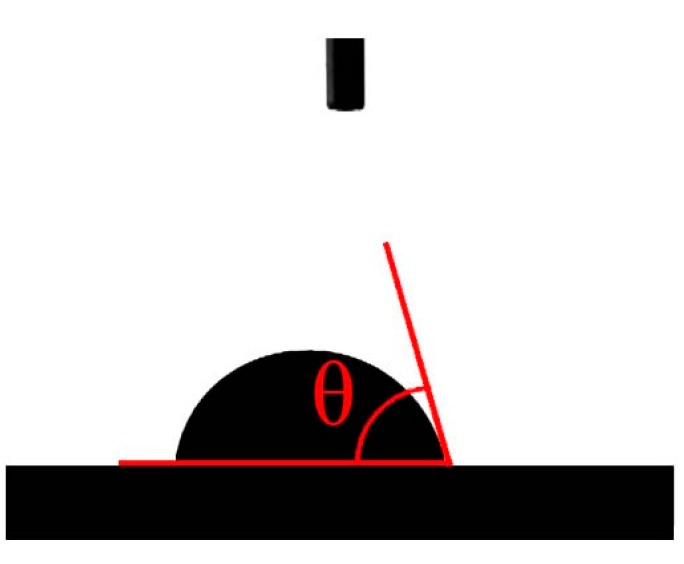
Schematic of contact angle testing

**Figure 6 materials-12-01796-f006:**
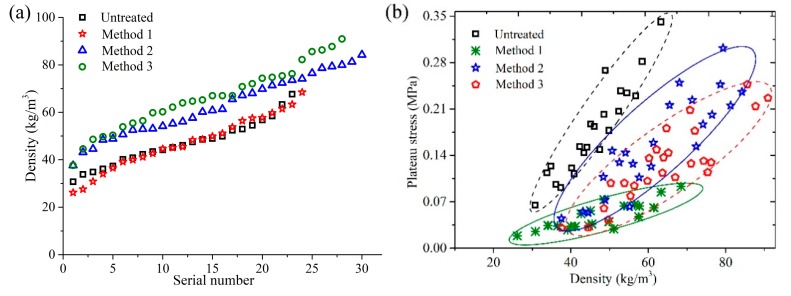
(**a**) Density of CL after treatment with the three methods; (**b**) plateau stress of CL after treatment with the three methods.

**Figure 7 materials-12-01796-f007:**
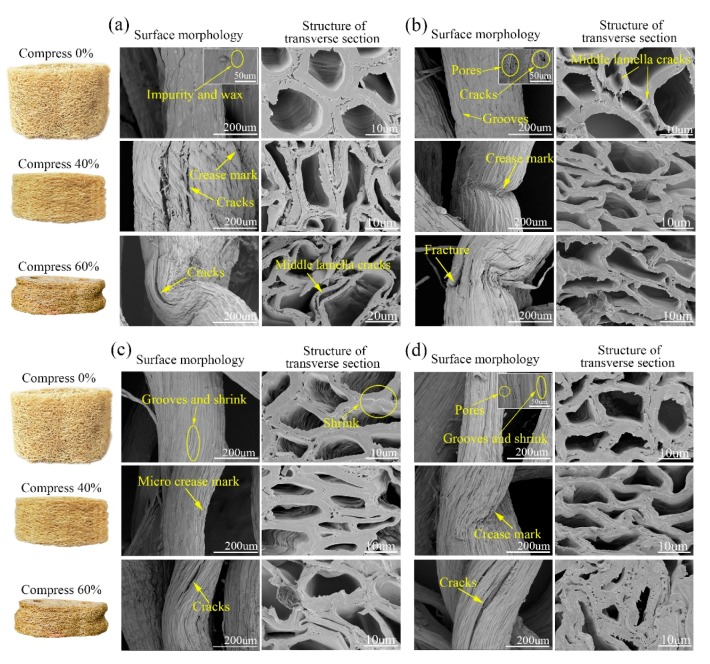
Surface morphology and microstructures of the inner luffa fiber bundles: (**a**) Untreated luffa; (**b**) luffa treated with Method 1; (**c**) luffa treated with Method 2; (**d**) luffa treated with Method 3.

**Figure 8 materials-12-01796-f008:**
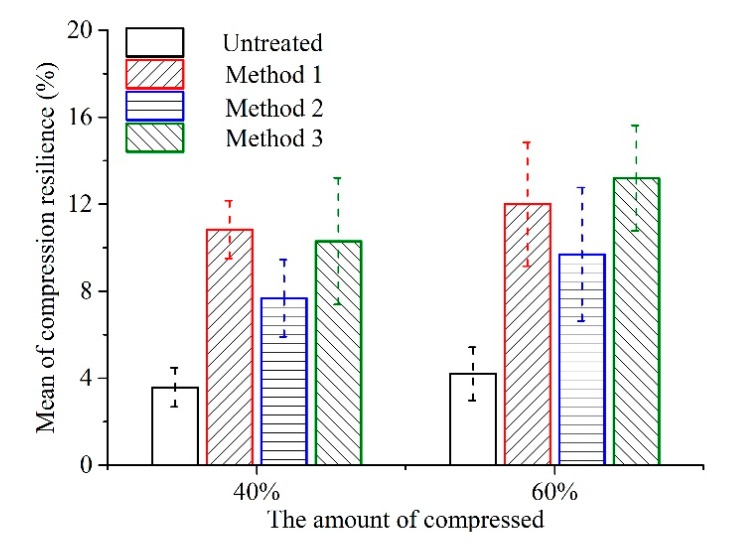
Mean of compression resilience of columned luffa.

**Table 1 materials-12-01796-t001:** The correlation coefficients of luffa density before and after treatment.

Parameters	Before and after Method 1 Treatment	Before and after Method 2 Treatment	Before and after Method 3 Treatment
Correlation coefficients	0.92	0.708	0.808

**Table 2 materials-12-01796-t002:** The mean and analysis of variance values of plateau stress for untreated and treated columned luffa.

Parameters	Untreated	Method 1	Method 2	Method 3	F-Value	P
Plateau stress (MPa)	0.175 ± 0.069	0.047 ± 0.021	0.157 ± 0.073	0.127 ± 0.058	19.124	0.000
A	7.302 × 10^−5^	8.731 × 10^−5^	6.931 × 10^−5^	5.139 × 10^−5^	-	-
B	2.027	1.626	1.862	1.87	-	-
R^2^	0.873	0.694	0.751	0.733	-	-

Note: The fitting equation of plateau stress is y = Ax^B^.

**Table 3 materials-12-01796-t003:** The mechanical properties of mattresses of untreated and treated columned luffa which were hot-pressed to smooth plateau region.

Samples	Compression Degree	*D_surface_* (MPa)	*D_core_* (MPa)	*D_bottom_* (MPa)	Hysteresis Loss Rate (%)	Compressive Deflection Coefficient
A11	40%	0.065 (0.017)	0.151 (0.021)	0.168 (0.024)	57.34 (1.43)	5.56 (0.13)
A12	40%	0.076 (0.011)	0.167 (0.052)	0.192 (0.024)	57.37 (10.3)	4.89 (0.82)
A13	40%	0.129 (0.023)	0.334 (0.092)	0.379 (0.034)	42.26 (1.70)	4.69 (0.78)
B11	40%	0.062 (0.006)	0.134 (0.010)	0.106 (0.007)	67.4 (2.30)	4.98 (0.13)
B12	40%	0.068 (0.004)	0.147 (0.010)	0.156 (0.007)	60.8 (2.96)	4.44 (0.11)
B13	40%	0.060 (0.002)	0.198 (0.011)	0.204 (0.023)	56.3 (1.52)	4.65 (0.44)
C11	40%	0.055 (0.006)	0.151 (0.002)	0.161 (0.014)	54.9 (0.84)	4.15 (0.10)
C12	40%	0.043 (0.005)	0.193 (0.004)	0.265 (0.014)	53.8 (0.12)	4.67 (0.18)
C13	40%	0.076 (0.001)	0.223 (0.027)	0.256 (0.032)	53.4 (1.22)	3.74 (0.14)
D11	40%	0.042 (0.008)	0.139 (0.005)	0.147 (0.031)	58.4 (0.59)	3.56 (0.08)
D12	40%	0.052 (0.002)	0.159 (0.012)	0.184 (0.036)	57.8 (0.77)	3.61 (0.72)
D13	40%	0.057 (0.001)	0.293 (0.009)	0.379 (0.012)	45.2 (0.52)	5.95 (0.37)

Note: The data is the mean of three repeated experiments, and the standard deviation of the three experiments is given in parentheses. A21, A22 and A23 refer to mattress samples filled with untreated columned luffa having a density range from 30 to 41 kg m^−3^, 42 to 50 kg m^−3^ and 51 to 65 kg m^−3^, respectively. B21, B22 and B23 refer to mattress samples filled with Method 1-treated columned luffa for which the original density range was similar to that of untreated LS. C21, C22 and C23 refer to mattress samples filled with Method 2-treated columned luffa with a similar original density range as untreated LS. D21, D22 and D23 refer to mattress samples filled with Method 3-treated columned luffa with a similar original density range as untreated LS.

**Table 4 materials-12-01796-t004:** The mechanical properties of mattresses of untreated and treated columned luffa that were hot-pressed to initial densification region.

Samples	Compression Degree	*D_surface_* (MPa)	*D_core_* (MPa)	*D_bottom_* (MPa)	Hysteresis Loss Rate (%)	Compressive Deflection Coefficient
A21	60%	0.048 (0.006)	0.102 (0.001)	0.171 (0.037)	56.55 (15.60)	8.92 (2.30)
A22	60%	0.061 (0.002)	0.137 (0.047)	0.181 (0.025)	51.64 (8.72)	11.17 (1.98)
A23	60%	0.068 (0.009)	0.178 (0.056)	0.271 (0.004)	54.23 (4.47)	12.18 (1.01)
B21	60%	0.040 (0.007)	0.110 (0.011)	0.164 (0.009)	56.4 (0.80)	13.79 (1.33)
B22	60%	0.051 (0.002)	0.126 (0.008)	0.183 (0.008)	52.3 (1.16)	10.63 (1.29)
B23	60%	0.049 (0.004)	0.132 (0.011)	0.207 (0.021)	54.2 (1.01)	10.77 (0.36)
C21	60%	0.044 (0.004)	0.109 (0.014)	0.141 (0.018)	51.1 (1.48)	9.55 (0.51)
C22	60%	0.040 (0.005)	0.145 (0.015)	0.231 (0.035)	45.3 (0.99)	11.63 (0.26)
C23	60%	0.058 (0.020)	0.172 (0.012)	0.240 (0.024)	45.8 (0.82)	8.88 (0.15)
D21	60%	0.030 (0.005)	0.088 (0.013)	0.120 (0.026)	52.9 (2.37)	8.73 (0.13)
D22	60%	0.038 (0.006)	0.110 (0.019)	0.146 (0.005)	53.3 (0.11)	6.54 (0.20)
D23	60%	0.040 (0.018)	0.118 (0.011)	0.171 (0.012)	47.4(1.11)	7.14 (0.12)

Note: The data is the mean of three repeated experiments, and the standard deviation of the three experiments is given in parentheses. A21, A22 and A23 refer to mattress samples filled with untreated columned luffa having a density range from 30 to 41 kg m^−3^, 42 to 50 kg m^−3^ and 51 to 65 kg m^−3^, respectively. B21, B22 and B23 refer to mattress samples filled with Method 1-treated columned luffa for which the original density range was similar to that of untreated LS. C21, C22 and C23 refer to mattress samples filled with Method 2-treated columned luffa with a similar original density range as untreated LS. D21, D22 and D23 refer to mattress samples filled with Method 3-treated columned luffa with a similar original density range as untreated LS.

**Table 5 materials-12-01796-t005:** Contact angle of four kinds of luffa.

Treatment Method	Untreated	Method 1	Method 2	Method 3	F-value	P
Contact angle θ	74.1 ± 2°	59.0 ± 4°	60.9 ± 2°	57.1 ± 1°	46.4	0.000

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
