# Peer review of "Effect of Chemical Treatments on the Properties of High-Density Luffa Mattress Filling Materials"

_materials, 2019, doi:10.3390/ma12111796_

Round 1

Reviewer 1 Report

This paper presents the effect of chemical treatments on the properties of high density luffa mattress filling materials. My main question for this research what are the difference between this work and the work of Chen which has been mentioned in lines 69. Both are using the same treatment methods with similar concentration.

The manuscript contains many brands of commercial companies

In Sec. 2.1.2, How did the authors measured the density.

In conclusion 3, what do you mean with good (line 407)

Author Response

Dear Professor,

On behalf of my co-authors, we appreciate you very much for your positive and constructive comments and suggestions on our manuscript entitled “Effect of chemical treatments on the properties of high-density luffa mattress filling materials” (ID: materials-474877). Those comments are all valuable and very helpful for revising and improving our paper, as well as the important guiding significance to our researches.

We have studied your comments carefully and have made revision which marked in yellow in the paper. We have tried our best to revise our manuscript according to the comments. Attached please find the revised version, which we hope meet with approval.

We would like to express our great appreciation to you for comments on our paper. The main corrections in the paper and the responses to your comments are as follows.

Thank you and best regards.

Yours sincerely,

Kaiting Zhang

Yuxia Chen

----------------------------------------------------------------------------------------

Responds to the comments:

Reviewer #1:

Comments and Suggestions for Authors

1) This paper presents the effect of chemical treatments on the properties of high density luffa mattress filling materials. My main question for this research what are the difference between this work and the work of Chen which has been mentioned in lines 69. Both are using the same treatment methods with similar concentration.

Response: Thanks for your valuable comments. Their research content was different. In Industrial Crops & Products 123 (2018) 341–352, the authors study the physical and mechanical properties of luffa fibers for the preparation of composite. Three chemical treatments were used to modify the LS fiber bundles. The results showed that the treatment with 10%NaOH-20%CH3COOH could significantly improve the mechanical. However, in this manuscript, we are investigated the variety properties of soften cylindrical luffa as potential mattress cushioning. Luffa is the matured dried fruit of towel gourd with a fibro-vascular system. Luffa fibers have been used for filling mattresses and pillows, but this usage destroys the natural networks of luffa, which causes collapse of mattresses/pillows and makes the mattress/pillow harder after being used for a period of time. Therefore, it is necessary to develop methods to make full use of the natural networks of luffa and prepare luffa-based mattress cushioning material.

2) The manuscript contains many brands of commercial companies

Response: Thanks for your valuable comments and suggestion. The raw material used in this paper, luffa, was provided by the company mentioned in line 95, so we add the name of the company to describe the source of the raw material. In order to avoid conflicts of interest, we have made modifications in the revised manuscript according to your suggestions. The detailed revision can be found in line 116.

3) In Sec. 2.1.2, How did the authors measured the density.

Response: Thanks for your valuable comments and suggestion. According to your suggestion, we add the process of calculating the density of the cylindrical luffa in the manuscript. The revised details can be found in line 121-139.

4) In conclusion 3, what do you mean with good (line 407)

Response: Thanks for your valuable comments. The good point here is that luffa can dissipate water quickly.

Thank you very much for your good comments and suggestions

Reviewer 2 Report

This paper evaluates the feasibility of different chemical treatment methods on the properties of  high-density luffa mattress filling materials. The manuscript is well organized, research work is properly designed, and the conclusions are supported by results. There are a few aspects which need clarification for enhanced clarity and readability.

I have appended major and minor comments below for Authors’ consideration.

1.     Why only three softening methods (5%NaOH-5%H2O2, 10%NaOH-20%CH3COOH and 18%NaOH-1.6%CO(NH2)2) were used to treat HD CL? The research novelty is unclear. The Authors have used the previously adopted pretreatment methods on luffa and determined the resulting properties. How do the Authors justify the contribution to existing knowledge base?

2.     Properties of raw luffa (as obtained from the manufacturer) should be tabulated separately. Also, the morphological features should be explained by providing photograph of the raw luffa as well.

3.     Line 134; please note that SEM requires complete drying of specimens prior to gold coating. Mention the details of drying, if any, in the text body.

4.     Figure 2 should better be revised. In my opinion, the needle image should be covered in the whole picture.

Author Response

Dear Professor,

On behalf of my co-authors, we appreciate you very much for your positive and constructive comments and suggestions on our manuscript entitled “Effect of chemical treatments on the properties of high-density luffa mattress filling materials” (ID: materials-474877). Those comments are all valuable and very helpful for revising and improving our paper, as well as the important guiding significance to our researches.

We have studied your comments carefully and have made revision which marked in yellow in the paper. We have tried our best to revise our manuscript according to the comments. Attached please find the revised version, which we hope meet with approval.

We would like to express our great appreciation to you for comments on our paper. The main corrections in the paper and the responses to your comments are as follows.

Thank you and best regards.

Yours sincerely,

Kaiting Zhang

Yuxia Chen

--------------------------------------------------------------------------------------

Responds to the comments:

Reviewer #2:

 1.     Why only three softening methods (5%NaOH-5%H2O2, 10%NaOH-20%CH3COOH and 18%NaOH-1.6%CO(NH2)2) were used to treat HD CL? The research novelty is unclear. The Authors have used the previously adopted pretreatment methods on luffa and determined the resulting properties. How do the Authors justify the contribution to existing knowledge base?

Response: Thanks for your valuable comments. The reason why we choose these three softening methods is that these methods are relatively common and we studied the effect of these methods on the central fiber bundle of the luffa previously. The results showed that three softening methods is not just change the characteristics of the surface topography, but also have effect on the mechanical properties of fiber bundles. Among them, NaOH-5%H2O2 could sharply decrease the mechanical properties of fiber bundles, the reduction of the tensile strength was 75.0%. The treatment with 10%NaOH-20%CH3COOH could significantly improve the mechanical properties of LS fiber bundles, the increase of the tensile strength was 121.3%. Luffa fibers have been used for filling mattresses and pillows, but this usage destroys the natural networks of luffa, which causes collapse of mattresses/pillows and makes the mattress/pillow harder after being used for a period of time. Therefore, it is necessary to develop methods to make full use of the natural networks of luffa and prepare luffa-based mattress cushioning material. In this manuscript, we are investigated the properties of softened cylindrical luffa as potential mattress cushioning. The revised details can be found in line 479-484

2.     Properties of raw luffa (as obtained from the manufacturer) should be tabulated separately. Also, the morphological features should be explained by providing photograph of the raw luffa as well.

Response: Thanks for your valuable comments and suggestions. The raw luffa is shown in Figure 2. In our manuscript, the cylindrical luffa is a 50 mm columnar cut from the raw luffa. So the properties of untreated cylindrical luffa can represent the properties of the raw material. Because untreated cylindrical luffa is also the object of our research, the untreated cylindrical luffa was studied in comparison with chemically treated cylindrical luffa for studying the softening effect.

3.     Line 134; please note that SEM requires complete drying of specimens prior to gold coating. Mention the details of drying, if any, in the text body.

Response: Thanks for your valuable comments and suggestions. The samples observed by SEM were middle layer fiber bundles taken from the cylindrical luffa. The cylindrical luffa was placed in a humidity chamber at 20℃ and 65% RH for 24 h, and then compressed or uncompressed. Then the fiber bundles were extracted from the inner layer of the cylindrical luffa. This fiber bundles can be scanned by SEM. Therefore, it can be considered that these luffa is treated by natural drying treatment. We have added drying method in line 183.

4.     Figure 2 should better be revised. In my opinion, the needle image should be covered in the whole picture.

Response: Thanks for your valuable comments and suggestions. We have revised Figure 2 according to your suggestion, using the full image. The revised details can be found in line 233.

Thank you very much for your good comments and suggestions

Reviewer 3 Report

1) The originality and the scientific value of the subject are good. 

Indeed, an important problem having direct applications is treated.

2) The Abstract in its current form is not sufficient. In particular, it should be supported in a more effective manner by the results obtained during research, because the first part which is read by Journal’s audience is Abstract and thus it should reflect the novelty and perform the main results. 

3) The Introduction Section in its current form is not adequate. 
In this context, I recommend the authors to add more literature review. There are many valuable works going on in this area. Besides, the differences/advantages of the present investigation compared to other literature works should be written out at the end of this Section in a much more detailed and comprehensive manner.

4) The presentation of the experimental work is very thorough. Process and prerequisites of sample preparation are clearly mentioned. However, the authors are kindly recommended to provide some further technical details about the laboratory equipment that they used to carry out their experiments.

5) The presentation and clarity of results and data are very good. Yet, the quality of Figure 3a,b is not in publication level. Moreover, this work would be more complete, in our opinion, if the experiments would be combined with a theoretical/analytical study. Further, is there any possibility for comparison with advanced computational methods (FEM, BEM)?

6) Logic and coherence are concrete and the clarity and quality of writing are sound.

7)  The Conclusions Section performs the findings of this work in a rather brief manner.

It should become more thorough and comprehensive. Also, I invite the authors to add a paragraph on the motives and prospects that this work provides for future research.

Overall, it is the reviewer’s opinion that this manuscript may be recommended for publication provided that the authors account for these critical remarks and revise the manuscript accordingly.

Author Response

Dear Professor,

On behalf of my co-authors, we appreciate you very much for your positive and constructive comments and suggestions on our manuscript entitled “Effect of chemical treatments on the properties of high-density luffa mattress filling materials (ID: materials-474877). Those comments are all valuable and very helpful for revising and improving our paper, as well as the important guiding significance to our researches.

We have studied your comments carefully and have made revision which marked in yellow in the paper. We have tried our best to revise our manuscript according to the comments. Attached please find the revised version, which we hope meet with approval.

We would like to express our great appreciation to you for comments on our paper. The main corrections in the paper and the responses to your comments are as follows.

Thank you and best regards.

Yours sincerely,

Kaiting Zhang

Yuxia Chen

-----------------------------------------------------------------------------------------

Responds to the comments:

Reviewer #3:

Comments and Suggestions for Authors

1) The originality and the scientific value of the subject are good. Indeed, an important problem having direct applications is treated.

 Response: Thanks for your valuable comments.

2) The Abstract in its current form is not sufficient. In particular, it should be supported in a more effective manner by the results obtained during research, because the first part which is read by Journal’s audience is Abstract and thus it should reflect the novelty and perform the main results. 

Response: Thanks for your valuable comments and suggestions. According to your suggestions, we have added the experimental results to enrich the summary content. The detailed content was in line 21-23 and in line 27-28.

3) The Introduction Section in its current form is not adequate. In this context, I recommend the authors to add more literature review. There are many valuable works going on in this area. Besides, the differences/advantages of the present investigation compared to other literature works should be written out at the end of this Section in a much more detailed and comprehensive manner.

Response: Thanks for your valuable comments and suggestions. According to your suggestion, we have added the literature review and highlights and the difference with other researches in the preface of the revised manuscript, the detailed content was in line 80-91 and in line 103-108.

4) The presentation of the experimental work is very thorough. Process and prerequisites of sample preparation are clearly mentioned. However, the authors are kindly recommended to provide some further technical details about the laboratory equipment that they used to carry out their experiments.

Response: Thanks for your valuable comments and suggestions. We have revised the manuscript according to your comments. The amend places have been highlighted.

5) The presentation and clarity of results and data are very good. Yet, the quality of Figure 3a, b is not in publication level. Moreover, this work would be more complete, in our opinion, if the experiments would be combined with a theoretical/analytical study. Further, is there any possibility for comparison with advanced computational methods (FEM, BEM)?

Response: Thanks for your valuable comments and suggestions. Based on your suggestion, we have revised Figure 3a.b. Thank you very much for your suggestions on the method of testing. We are not familiar with these two methods, unable to guarantee the accuracy of the results. We hope to use these methods in future research. We have added the detailed steps of the measurement method of the density of the cylindrical luffa in the revised manuscript, the detailed content was in line 122-144.

6) Logic and coherence are concrete and the clarity and quality of writing are sound.

Response: Thanks for your valuable comments.

7)  The Conclusions Section performs the findings of this work in a rather brief manner. It should become more thorough and comprehensive. Also, I invite the authors to add a paragraph on the motives and prospects that this work provides for future research.

Response: Thanks for your valuable comments and suggestions. According to your suggestions, we have revised the conclusion section, and added the purpose and prospect of the paper. The revised details can be found in line 462-464 and line 479-484.

Thank you very much for your good comments and suggestions

Reviewer 4 Report

Title: Effect of chemical treatments on the properties of high-density luffa mattress filling materials.

Authors: Kaiting Zhang and colleagues

Overall assessment:

The authors investigate several mechanical properties of HD luffa mattress filling materials. The topic adopted in this manuscript may be interesting for several researchers. Nevertheless, there are many casualties in the descriptions; therefore, there is a concern that many readers cannot appreciate the essence of the study. The authors must improve the manuscript by careful and objective readings into that many readers, including undergraduate students, can understand the essence of the study easily and approprieately.

Specific comments:

1. When seeing the introduction section, it is difficult to understand the difference between the low-density luffa and high-density luffa is ambiguous. The ranges of the corresponding densities must be denoted in the introduction section.

2. It is difficult to understand how to determine the densities listed in the lines 102 and 103 because of the lack of the definition of the volume used for the determination of the density. Which was the volume used here gross or net one? Even if the measurement method was described in a previous paper in details, this issue must be denoted in the manuscript.

3. In Line 105: Methods 1, 2, and 3 must be defined corresponding to the treatments.

4. Line 133: “Hitachi” is mistyped as “Hatachi”.

5. There are no descriptions how to measure the strain in the compression test. The apparatus for measuring the strain must be denoted.

6. It is preferable to demonstrate the stress-strain relation during the compression test to clarify the meanings of Eqs. (3)-(7). I think that the stress-strain diagram will enhance the precise understanding of the meaning of various parameters obtained during the compression test such as those in Eqs. (3)-(7), plateaus stress, etc.

7.Line 155: What is “(75+1)%”? Is it 76%?

8. There are no descriptions on Figs. 3 and 4 and Tables 1 and 2 before the first appearance. The locations of the figures and tables must be rearranged.

9. Lines 194 and 195: Perhaps “Figure 3” is mistyped as “Figure 4”.

10. The label of the horizontal axis of Fig. 3(a) is missing.

11. The significant figures of the values listed in Table 5 must be checked.

12. Perhaps many readers would like to know which treatment is consequently most effective.

Recommendation

Thorough revisions are required and the authors must prepare the revised version as well as the point-by-point response to the comments described above. The casual descriptions often prevent the readers from precise understandings of the essence of this study; therefore, the authors must improve the manuscript by careful and objective readings. I’d like to receive a revised version in due course.

Author Response

Dear Professor,

On behalf of my co-authors, we appreciate you very much for your positive and constructive comments and suggestions on our manuscript entitled “Effect of chemical treatments on the properties of high-density luffa mattress filling materials” (ID: materials-474877). Those comments are all valuable and very helpful for revising and improving our paper, as well as the important guiding significance to our researches.

We have studied your comments carefully and have made revision which marked in yellow in the paper. We have tried our best to revise our manuscript according to the comments. Attached please find the revised version, which we hope meet with approval.

We would like to express our great appreciation to you for comments on our paper. The main corrections in the paper and the responses to your comments are as follows.

Thank you and best regards.

Yours sincerely,

Kaiting Zhang

Yuxia Chen

-----------------------------------------------------------------------------------

Responds to the comments:

Reviewer #4:

Specific comments:

 1. When seeing the introduction section, it is difficult to understand the difference between the low-density luffa and high-density luffa is ambiguous. The ranges of the corresponding densities must be denoted in the introduction section.

Response: Thanks for your valuable comments and suggestions. We have added the density range of low-density luffa in line 67. The density range of high-density Luffa has been written out in line 68.

2. It is difficult to understand how to determine the densities listed in the lines 102 and 103 because of the lack of the definition of the volume used for the determination of the density. Which was the volume used here gross or net one? Even if the measurement method was described in a previous paper in details, this issue must be denoted in the manuscript.

Response: Thanks for your valuable comments and suggestions. We are very sorry that we lacked the definition of the volume used for the determination of the density. According to your suggestions, we have added the detailed steps of the measurement density in the revised manuscript, in line 122-140.

3. In Line 105: Methods 1, 2, and 3 must be defined corresponding to the treatments.

Response: Thanks for your valuable comments. It was our mistake. We have added Methods 1, 2 and 3 in the revised manuscript, the detailed content was in line 145-146.

4. Line 133: “Hitachi” is mistyped as “Hatachi”.

Response: Thanks for your valuable comments. We have revised it in the revised manuscript, in the line 182.

5. There are no descriptions how to measure the strain in the compression test. The apparatus for measuring the strain must be denoted.

Response: Thanks for your valuable comments and suggestions. We are very sorry for our incorrect the subheading, we have revised it in the revised manuscript. In the manuscript, we used the mechanical test machine to collect the force and displacement in the compression process of the cylindrical luffa, and then used the force ratio to the area to obtain the stress, and the displacement ratio to the original height to obtain the strain. The original height and area of the cylindrical luffa were obtained by measuring the density of the cylindrical luffa.

6. It is preferable to demonstrate the stress-strain relation during the compression test to clarify the meanings of Eqs. (3)-(7). I think that the stress-strain diagram will enhance the precise understanding of the meaning of various parameters obtained during the compression test such as those in Eqs. (3)-(7), plateaus stress, etc.

 Response: Thanks for your valuable comments and suggestions. According to your suggestion, we added the compressive stress-strain curve in the revised manuscript, and illustrate the computation process of formula (3) - (7) and densification strain in the figure, the detailed content was in line 206-207.

7.Line 155: What is “(75+1)%”? Is it 76%?

Response: Thanks for your valuable comments. We are very sorry for our incorrect writing. Line 209 should be (75±1) %.

8. There are no descriptions on Figs. 3 and 4 and Tables 1 and 2 before the first appearance. The locations of the figures and tables must be rearranged.

Response: Thanks for your valuable comments and suggestions. We have added the descriptions of Figure 3 in line 242-243, the descriptions of Figure 4 in line 246-247, the descriptions of Table 1 in line 246, the descriptions of Table 2 in line 277-278.

9. Lines 194 and 195: Perhaps “Figure 3” is mistyped as “Figure 4”.

Response: Thanks for your valuable comments. We are very sorry for our incorrect writing. We have revised the figure number in the revised manuscript, and checked other figure/table numbers in the manuscript carefully. The revised places have been highlighted.

10. The label of the horizontal axis of Fig. 3(a) is missing.

Response: Thanks for your valuable comments. We have added the name of horizontal axis in the revised manuscript, the detailed content was in line 237.

11. The significant figures of the values listed in Table 5 must be checked.

Response: Thanks for your valuable comments and suggestions. We are very sorry for our incorrect writing. We have corrected them in the revised manuscript, in line 431 and line 440.

12. Perhaps many readers would like to know which treatment is consequently most effective.

Response: Thanks for your valuable comments. We have written in the abstract and conclusion that Method 3 (18% NaOH-1.6% CO(NH2)2) is the most effective method. After the treatment by Method 3, the luffa fiber bundle was intact, and the compression resilience of the cylindrical luffa was obviously increased. Therefore, this method can effectively reduce the firmness of mattress filling materials, and also improve the uniformity and wettability of cylindrical luffa.

Thank you very much for your good comments and suggestions

Round 2

Reviewer 2 Report

I recommend that the paper be accepted. However, there are a lot of language issues. I suggest to get the manuscript proofread by a native English speaker expert in this field.

Reviewer 3 Report

The authors accounted for my critical remarks and made adequate improvements.

Thus, I am satisfied with the manuscript it its current form and I recommend it for publication.  

Reviewer 4 Report

The manuscript is adequately revised; therefore, I'd like to recommend the present version to be published as it is.